# Identification of Indicator Genes for Agar Accumulation in *Gracilariopsis lemaneiformis* (Rhodophyta)

**DOI:** 10.3390/ijms25094606

**Published:** 2024-04-23

**Authors:** Zheng Li, Mengge Tu, Feng Qin, Guangqiang Shui, Di Xu, Xiaonan Zang

**Affiliations:** Key Laboratory of Marine Genetics and Breeding, Ministry of Education, Ocean University of China, Qingdao 266003, China; lz6780@stu.ouc.edu.cn (Z.L.); 21220611168@stu.ouc.edu.cn (M.T.); 11220611054@stu.ouc.edu.cn (F.Q.); 21200631068@stu.ouc.edu.cn (G.S.)

**Keywords:** *Gracilariopsis lemaneiformis*, agar biosynthesis, indicator genes, qPCR

## Abstract

Agar, as a seaweed polysaccharide mainly extracted from *Gracilariopsis lemaneiformis*, has been commercially applied in multiple fields. To investigate factors indicating the agar accumulation in *G. lemaneiformis*, the agar content, soluble polysaccharides content, and expression level of 11 genes involved in the agar biosynthesis were analysed under 4 treatments, namely salinity, temperature, and nitrogen and phosphorus concentrations. The salinity exerted the greatest impact on the agar content. Both high (40‰) and low (10‰, 20‰) salinity promoted agar accumulation in *G. lemaneiformis* by 4.06%, 2.59%, and 3.00%, respectively. The content of agar as a colloidal polysaccharide was more stable than the soluble polysaccharide content under the treatments. No significant correlation was noted between the two polysaccharides, and between the change in the agar content and the relative growth rate of the algae. The expression of all 11 genes was affected by the 4 treatments. Furthermore, in the cultivar 981 with high agar content (21.30 ± 0.95%) compared to that (16.23 ± 1.59%) of the wild diploid, the transcriptional level of 9 genes related to agar biosynthesis was upregulated. Comprehensive analysis of the correlation between agar accumulation and transcriptional level of genes related to agar biosynthesis in different cultivation conditions and different species of *G. lemaneiformis*, the change in the relative expression level of glucose-6-phosphate isomerase II (*gpiII*), mannose-6-phosphate isomerase (*mpi*), mannose-1-phosphate guanylyltransferase (*mpg*), and galactosyltransferase II (*gatII*) genes was highly correlated with the relative agar accumulation. This study lays a basis for selecting high-yield agar strains, as well as for targeted breeding, by using gene editing tools in the future.

## 1. Introduction

The inner cell wall matrix of red algae from the families of Gelidiaceae, Pterocladiaceae, Gelidiellaceae, and Gracilariaceae contains agar, which serves as the major extracellular polysaccharides [1,2,3,4]. In algae, the cell wall polysaccharide plays an essential role in adaptation to the marine environment, such as providing flexibility to cope with strong ocean currents and serving as a barrier against osmotic pressure and pathogens [5]. Additionally, as a marine natural material, the hydrophilic colloid caters to diverse applications in nutraceutical fields, pharmaceuticals, and biotechnology industries [6,7]. Agar remains as a gel under ambient temperatures [8]. Even at a low density of 1%, agar remains a relatively stable jelly, and, because of this characteristic, it is considered valuable in the industry. Globally, agar production has escalated from 9600 tons in 2009 to 15,000 tons in 2019 [9,10]. Nevertheless, the supply shortage of agar remains [11,12]. In 2003, the largest amount of agar in the world was produced from *Gracilaria* (53%), followed by *Gelidium* (44%) [4]. By 2015, the proportion of *Gracilaria* in the collected agarophytes reached 91%, whereas that of *Gelidium* continued to decline, partly because *Gelidium* still cannot be cultivated [13]. *Gracilariopsis lemaneiformis*, which has been successfully cultivated in China, is among the main sources of agar. *G. lemaneiformis* produces a superior yield of agar having high gel strength [14,15]. Cultivars 981 and 2007 were bred with high agar content and strong heat resistance have shown significant advantages in cultivation and have become the main cultivated variety [16]. The increasing industrial demand for *G. lemaneiformis* urgently requires the cultivation of new breeds with higher agar content, which requires understanding the biosynthesis pathway of agar.

Agar is a mixture of gelling polysaccharide comprising agarose as the main component and agaropectin, which is agarose with different conformations and side chain substitutions, including sulfate ester, methoxyl group, and pyruvate ketal [17]. The current hypothetical agar biosynthesis pathway was deduced through chemical analysis in vascular plants and red seaweeds [18]. The proposed pathway starts from fructose-6-phosphate, followed by the glycosylation of alternating UDP-D-galactose and GDP-L-galactose to form the agar precursor chain. Then, the side chain is modified and the polymer is transferred to the cell wall matrix [19]. Based on the metabolic pathway, several studies have reported the relationship between the expression level of related genes and agar content in some Gracilariaceae species [20]. Environmental conditions were found to affect the expression of the agar biosynthesis-related genes, which then affects the agar content. The agar yield of algae is generally elevated in summer, which is correlated with the increase in water temperature, and when the alga is grown under altered salinity, but nitrogen enrichment in the seawater may reduce agar production [21]. The agar content of *G. lemaneiformis* increased considerably in hypertonic, hypotonic, nitrogen-limitation, and phosphorus-limitation cultures [20,22]. Fundamentally, agar is accumulated because of an increase in the expression abundance of relevant genes. Hence, the correlation between genes and the agar content under external interference conditions should be confirmed, which may be beneficial for screening for molecular markers directly related to agar content.

In the breeding of *G. lemaneiformis*, how to screen seedlings with high agar content is crucial. The method existing for determining the agar yield is laborious, time-consuming, and requires relatively high amounts of raw materials [23]. So, it is difficult to detect the agar or other metabolite content in the early stage due to the small number of seedlings, which is non-conducive to the screening of a high-yield agar strain from numerous samples in the breeding. Many studies have shown that the accumulation of agar is due to an increase in the abundance of related gene expression. Transcription level detection is an efficient and fast method for detecting gene expression with a small amount of sample. Thus, a convenient and accurate approach involving the use of molecular markers is urgently required for the rapid screening of commercially potential strains [21].

According to the reports, nine enzymes are mainly involved in agar biosynthesis. Accordingly, 11 related genes have been cloned, namely glucose-6-phosphate isomerase I (*gpiI*), glucose-6-phosphate isomerase II (*gpiII*), phosphoglucomutase (*pgm*), UDP-glucose pyrophosphorylase (*ugp*), galactose-1-phosphate uridylyltransferase (*galt*), mannose-6-phosphate isomerase (*mpi*), phosphomannomutase (*pmm*), mannose-1-phosphate guanylyltransferase (*mpg*), GDP-mannose-3,5’-epimerase (*gme*), galactosyltransferase I (*gatI*), and galactosyltransferase II (*gatII*). In this study, the relationship between the agar accumulation, the soluble polysaccharides content, and the expression levels of the 11 agar biosynthesis-related genes was cross-verified in *G. lemaneiformis* cultivated under four external treatment conditions (salinity, temperature, and nitrogen and phosphorus concentrations) and in different varieties and generations of *G. lemaneiformis* (cultivar 981, wild female gametophyte, and wild diploid), so as to screen the most relevant marker for a high agar yield. These findings will enrich our knowledge about the cell wall polysaccharides biosynthesis at the molecular level and offer vital information for the future selection of strains with high agar yield and the artificial regulation of agar accumulation in algae.

## 2. Results

### 2.1. Growth Assessment

Under four treatments, *G. lemaneiformis* exhibited different relative growth rates (RGR) during the 15-day cultivation period (Figure 1), which indicated that these treatments had an intervention effect on the algal physiology. Although the algal RGR was the highest in 10‰ salinity in the first 6 days compared with other salinity levels, the RGR fluctuated markedly in 10‰ salinity (standard deviation: 1.36) and exhibited a gradual decreasing trend (from 4.58 ± 0.27%/day for 3 days to 1.22 ± 0.01%/day for 15 days, *p* < 0.05, Figure 1a), and then at the 15th day, it was close to the RGR of 20‰ salinity. For the 15-day period, the RGR values at 10‰ salinity (2.17 ± 1.35%/day) and 20‰ salinity (1.55 ± 0.55%/day) were higher than that at 30‰ salinity (0.81 ± 0.34%/day) with significant difference (*p* < 0.05). And the algae had a lower RGR in 40‰ salinity (0.24 ± 0.12%/day). This indicated that appropriate low salinity promoted *G. lemaneiformis* growth during the short-term cultivation period, whereas high salinity inhibited algal growth.

The algal RGR at 20 °C (2.56 ± 0.46%/day) was significantly higher than that at 5 °C (0.43 ± 0.13%/day), 10 °C (1.18 ± 0.55%/day), and 30 °C (1.75 ± 0.51%/day) (Figure 1b). The high temperature of 35 °C exceeded the temperature range that *G. lemaneiformis* can only withstand temporarily.

Compared with the N concentration of 6 μM (1.55 ± 0.11%/day), the 300 μM (2.19 ± 0.08%/day) and 900 μM (2.02 ± 0.12%/day) N concentration significantly increased the algal RGR (Figure 1c). Under laboratory conditions, 6 μM N concentration was unfavorable for *G. lemaneiformis* growth. A 30 μM N concentration could satisfy the growth of *G. lemaneiformis* in a short time(<6 days), and there was no significant difference in RGR between the 300 μM N concentration and 900 μM N concentration in the initial 6 days. But, with the extension of culture time, the RGR of the 30 μM N concentration decreased significantly, which was significantly lower than 300 μM N concentration and 900 μM N concentration, while 300 μM and 900 μM had no significant difference in 15 days, indicating that a N concentration above 300 μM was sufficient for *G. lemaneiformis*. Therefore, appropriate supplementation of nitrogen sources is beneficial for *G. lemaneiformis* growth in cultivation.

When the phosphorus concentration increased, the algal growth rate showed an increasing trend (Figure 1d). However, a significant difference was not noted among the effect of 0.5 μM (1.23 ± 0.16%/day), 1.5 μM (1.34 ± 0.28%/day), and 30 μΜ (1.49 ± 0.27%/day) P concentrations on algal RGR especially in the first 6 days. With the prolongation of culture time, 30 μM P concentration showed a certain growth-promoting effect.

### 2.2. Agar Content

Compared with the initial agar content, the agar content of algae cultured at 10‰, 20‰, and 40‰ salinity for 15 days significantly increased by 2.59%, 3.00%, and 4.06%, respectively (Figure 2a). Under 30‰ salinity (normal condition), the algal agar content exhibited no significant change, suggesting that both low and high salinity promoted agar accumulation in *G. lemaneiformis*. Compared with the initial agar content, the agar content of algae cultured at 5 °C significantly decreased by 2.71%, while the agar content of algae cultured at 30 ° C significantly increased by 3.40% (Figure 2b). No significant change was observed in the agar content of algae under the temperature pressure of 10 °C and the normal temperature of 20 °C. The N concentration has little effect on the agar content of *G. lemaneiformis*, with no significant change observed in the agar content of algae at N concentrations of 6 μM, 30 μM, 300 μM, and 900 μM (Figure 2c). Compared with the initial agar content, the agar content of the algae at 0.5 μM P concentration significantly increased by 3.66% (Figure 2d). However, the agar content exhibited no significant change at P concentrations of 1.5 μM and 30 μM. These results indicated that the effect of N and P concentrations on agar content was not as significant as that on growth. A relatively low-nutrient environment may be beneficial for agar biosynthesis in *G. lemaneiformis*. Additionally, compared with temperature, nitrogen, and phosphorus treatments, salinity stress can more effectively promote agar accumulation.

### 2.3. Soluble Polysaccharides Content

The soluble polysaccharides content of algae was significantly lower in the 10‰ salinity group than that of the other salinity groups at 6 days (Figure 3a). At 15 days, the soluble polysaccharides content was significantly lower in the 10‰ and 20‰ groups than in the 30‰ group and significantly lower in the 10‰ group than in the 40‰ group. This may be related to rapid algal growth at 10‰ and 20‰ salinity, resulting in lower soluble polysaccharides for storage. The soluble polysaccharides content at 40‰ salinity was also not higher than that at 30‰ salinity, which may be because of the impact of high salinity on algal growth and metabolism. At 15 days, the soluble polysaccharides content of algae was significantly higher in the 5 °C, 10 °C, and 30 °C groups than in the 20 °C group (Figure 3b). The high soluble polysaccharides content of *G. lemaneiformis* at low or high temperatures may be related to the slow algal growth and the accumulation of polysaccharides and other substances for improving their anti-stress ability. The treatments with both high and low N concentrations promoted the soluble polysaccharides content of *G. lemaneiformis*, especially the high N concentration (Figure 3c). At 15 days, the soluble polysaccharides content was significantly higher in the high N concentration group than that of the low N concentration groups. Thus, under N-rich conditions, the soluble polysaccharides content of *G. lemaneiformis* may increase with biomass accumulation. However, under different P concentration treatments (Figure 3d), the soluble polysaccharides content of *G. lemaneiformis* exhibited no significant change.

### 2.4. The Transcription Level of 11 Genes in G. lemaneiformis under Four Treatments

The expression of *gatII*, *ugp*, *mpg*, *mpi*, *galt*, and *gme* was upregulated, but that of *pmm*, *pgm*, and *gatI* was downregulated at 10‰ and 20‰ (Figure 4a). *gpiI* and *gpiII* expression was upregulated at 10‰ but was downregulated at 20‰. In general, the expression of all genes was upregulated at 40‰. The results suggested that salinity stress promoted the expression of most genes associated with agar biosynthesis.

The expression of *pmm*, *gme*, and *pgm* was upregulated, but that of *gatI*, *gatII*, *galt*, *mpi*, *mpg*, *gpiI*, *gpiII*, and *ugp* was downregulated at 5 °C, which indicated that low temperature inhibited the expression of most genes (Figure 4b). When algae grew at 10 °C, the expression of *gatI*, *gatII*, *galt*, and *mpi* was still downregulated, but that of *pmm*, *gme*, *mpg*, *gpiI*, *gpiII*, *pgm*, and *ugp* was upregulated. The expression of *gatI*, *gatII*, *galt*, *mpi*, *pmm*, *mpg*, *gpiI*, and *gpiII* was upregulated, but that of *gme*, *pgm*, and *ugp* was downregulated at 30 °C. Additionally, the expression of all genes was upregulated when algae were cultured at 35 °C. This indicated that high temperatures enhanced the expression of agar biosynthesis-related genes.

The expression of *gpiI*, *gpiII*, *gatI*, *gatII*, *ugp*, *mpg*, *galt*, and *gme* was upregulated, but that of *pgm*, *mpi*, and *pmm* was downregulated at the 6 μM N concentration (Figure 4c). The expression of *gatII*, *ugp*, and *pgm* was upregulated, but that of other genes was basically downregulated at the 300 μM N concentration. The expression of *gpiI*, *gpiII*, *gatI*, *gatII*, *ugp*, *pgm*, *galt*, and *gme* was downregulated, but that of *mpi*, *pmm*, and *mpg* was upregulated at the 900 μM N concentration. This revealed that the expression of different genes varied under different nitrogen concentrations. Under high or low nitrogen conditions, the expression of some genes was upregulated.

The expression of *mpg*, *mpi*, *gatII*, *gpiII*, *galt*, *gpiI*, and *pgm* was upregulated, whereas that of *gatI*, *pmm*, *gme*, and *ugp* was downregulated at the 0.5 μM P concentration (Figure 4d). However, *pmm*, *gme*, *mpi*, and *gatI* expression was upregulated, whereas the expression of other genes was downregulated at the high P concentration of 30 μM. This revealed that similar to the nitrogen concentration, under high or low phosphorus conditions, the expression of some genes was upregulated. The low P concentration treatment might stimulate the expression of more genes associated with agar biosynthesis.

Under salinity (10‰, 20‰, and 40‰), temperature (5 °C and 30 °C) stress, and low P (0.5 μM P) treatments, the agar content in *G. lemaneiformis* changed significantly, which was generally consistent with the expression changes of the *gpiI*, *gpiII*, *galt*, *mpi*, *mpg*, and *gatII* genes.

### 2.5. The Variation of Relationship between Agar Content and Gene Transcription Level in Different Varieties and Generations of G. lemaneiformis

To validate the relationship between the transcription levels of genes and agar content, different varieties and generations of *G. lemaneiformis* were used for analysis. The cultivar 981 is the major culture strain with a higher agar content than that of the wild type (19%), and the mature female gametophytes generally have a higher agar content than that of mature diploids. After being cultivated (culture conditions as in Section 4.1) for 7 days, the agar content of the cultivar 981, the wild female gametophyte, and the wild diploid of *G. lemaneiformis* were 21.30 ± 0.95%, 18.17 ± 1.57%, and 16.23 ± 1.59%, respectively (Figure 5a). Compared with the wild diploid, the agar content of 981 was significantly increased by 5.07%, while the agar content of the wild female gametophyte was slightly higher (1.94%) without the significant difference.

Compared to the wild diploid, except for the *gatI*, and *gme*, the *gpiI*, *gpiII*, *pgm*, *ugp*, *galt*, *mpi*, *pmm*, *mpg*, and *gatII*, had the highest transcription level in the cultivar 981 (Figure 5b), which was consistent with the high agar content of it. All six genes, the *gpiI*, *gpiII*, *galt*, *mpi*, *mpg*, and *gatII*, screened in the 2.4 results were upregulated in expression, indicating that these genes may serve as indicators of agar content.

For the wild female gametophytes, the expression of *gpiI*, *galt*, *mpi*, and *gme* was upregulated, while *pgm*, *gatI*, *gatII*, and *mpg* were downregulated for expression (Figure 5b), which may have resulted in the agar content of female gametophytes only being marginally higher than that of diploid, and did not show a significant difference.

### 2.6. The Relationship between Agar Content and Gene Transcription Level

According to the results of Section 2.4, under the same treatment condition at different times, whether on day 1, day 2, day 3, day 6, or day 9, the transcriptional level of the same gene showed consistent trends, and tended to be stable and significant on days 6 to 9, indicating that the genes undergo relatively stable changes with environmental changes. The results of Section 2.3 showed that the agar content also changed with the treatments. The trend between agar content and gene expression levels was further confirmed by the results of Section 2.5. These genes may be key genes for agar biosynthesis. Therefore, the correlation analysis between the gene's transcription level and agar content was conducted under different conditions to find the genes with the greatest relationship with agar content.

Based on the comprehensive calculation of C-value based on gene transcription levels for 1–9 days, combined with the relative accumulation of agar (RAA), the Pearson correlation analysis was performed. Table 1 presents the Pearson correlation coefficient (PCC). The C-values of the *gpiII*, *mpi*, *mpg*, and *gatII* were strongly and positively correlated with the RAA values, and their PCC was 0.925, 0.776, 0.901, and 0.856, respectively (Table 1 and Figure 6).

## 3. Discussion

Agar is among the most critical economic traits for agarophytes, and a huge supply–demand gap exists in the market. Therefore, acquiring agarophytes with higher agar yield and superior gelling properties has been desirable. The agar yield is not only related to the agarophyte species but can also be altered by changes in environmental factors, including cultivation salinity, temperature, and nitrogen and phosphorus concentrations [25]. Bird reported that the agar yield of the *Gracilaria* sp. strain G-16 was higher under long-term (4 weeks) low-salinity conditions (17‰) than at 33‰ salinity [26]. Chang verified that the agar content of *G. lemaneiformis* grown at low salinity (17‰) for 2 weeks apparently increased compared with that at 33‰ salinity (control group) [23]. The agar yield of *Gracilaria changii* grown at low salinity (10‰) was not significantly different from that grown at normal salinity (30‰). The agar yield of *Gracilaria changii* grown at high salinity (50‰) was more than 1.5-fold higher than that grown at normal salinity [27]. In this study, agar accumulation in *G*. *lemaneiformis* significantly increased at low salinity (10‰ and 20‰) and high salinity (40‰) compared with that at 30‰ salinity, especially at high salinity (Figure 2a). These results were consistent with the changes in the transcription levels of the genes. In particular, at the high-salinity concentration, the transcription levels of almost all the 11 agar biosynthesis-related genes increased. These results suggested that osmotic pressure might affect the agar content by inducing algae to synthesize more agar to increase cell wall strength, so as to combat the danger of cell rupture. Therefore, *G*. *lemaneiformis*, which can grow normally in a low or high-salinity environment for a long period, may have a high agar content.

The agar yield of agarophytes is related to water temperature [25]. The agar content of *Gracilaria bursa-pastoris* was positively correlated with temperature (r = 0.94; *p* < 0.01) [4]. Friedlander found that the agar yield of *Gracilaria conferta* cultured in tanks was positively correlated with water temperature [28]. In Chen’s study, the agar content of *G. lemaneiformis* cultured under heat stress (28 °C) for 11 days increased compared with that under the control condition (23 °C), but this difference was non-significant [29]. For *Gelidium* coulteri (with an optimal growth temperature of 25–27 °C), within the temperature range of 5–31 °C, agar accumulation in the algae gradually increased as the temperature increased [30]. However, within the temperature range of 10–30 °C, the agar content of *Gracilaria sordida* decreased with an increase in temperature [31]. In the present study, the agar content of *G*. *lemaneiformis* significantly decreased when cultured at a low temperature (5 °C) for 15 days. This may be because various physiological activities of algae were inhibited in the extremely low-temperature environment, thereby leading to the flow of carbohydrates for the synthesis of more crucial substances to maintain algal survival. Moreover, agar accumulation in *G*. *lemaneiformis* cultured at 20 °C for 15 days was higher than that under 10 °C, and the agar content was even higher under 30 °C with a significant difference. These were basically consistent with the changes in the transcription levels of almost all the genes, which were significantly improved at a high temperature, and most agar biosynthesis-related genes were generally inhibited at a low temperature.

Macronutrients such as nitrogen and phosphate are not only essential for seaweed growth but may also affect agar production by agarophytes. In general, the agar content of *Gracilaria* and *Gelidium* species is negatively correlated with the nitrogen level of the growth environment [31]. The agar content of *G*. *lemaneiformis* under N-limitation (normal seawater concentration) cultivation for 3 weeks increased significantly compared with that under normal cultivation (in Provasoli medium) [20,29]. Similar to previous studies, the present study found that the high nitrogen concentration (300 and 900 μM) did not promote agar accumulation. However, agar accumulation also exhibited no significant change at the 6 μM and 30 μM N concentrations, which may be attributable to the too-low N concentration, which is non-conducive to the living state of algal cells and agar accumulation. Similar results were also found in the culture experiment with different P concentrations. Studies of Hu proved that the agar content of *G*. *lemaneiformis* under P-limitation (normal seawater P concentration) cultivation for 3 weeks increased significantly compared with that under normal cultivation (in Provasoli medium) [20]. In the present study, agar accumulation in *G*. *lemaneiformis* increased significantly when cultured at the 0.5 μM P concentration for 15 days. However, no significant change was noted at the 1.5 and 30 μM P concentrations. In the transcriptional results of genes, the transcription levels of not all of the genes changed with a change in the nitrogen and phosphorus concentrations. However, changes in the expression of some genes were consistent with the changes in the agar content.

The soluble polysaccharides are crucial osmotic regulatory substances that can maintain the osmotic pressure of the algal cells, stability, and structure and function of biological macromolecules [32]. Under different treatments, the change in the content of agar, a colloid polysaccharide, may be correlated with the soluble polysaccharide content in the agarophytes. Compared with the control group, the soluble polysaccharides content decreased significantly at low salinity in *G*. *lemaneiformis*, but a significant change was not observed at high salinity. Under salinity stress, the variation in agar and soluble polysaccharides content was not completely consistent, possibly because most carbohydrate accumulates during the agar biosynthesis to increase cell strength. At low temperatures, the soluble polysaccharides content increased significantly, which may be because algae need more soluble polysaccharides to improve intracellular mobility and osmotic pressure, which thus adversely affects agar biosynthesis. However, the soluble polysaccharides and agar content increased at a high temperature, which may be related to the more active activity of the cells in some respects under high-temperature conditions. The soluble polysaccharides content at 300 and 900 μM N concentrations increased slightly, but the change was non-significant, which may be due to active cellular metabolism. At low N and P concentrations, the soluble polysaccharides content improved, to a certain extent, which is similar to the agar content. This may be related to the flow of carbon and nitrogen when a change occurs in nutrient concentrations. In short, the soluble polysaccharides are related to the growth, stress resistance, and cell state of algae, and their content varies considerably. The agar is mainly involved in the resilience and cell strength of algae, and the change in agar content was more stable than those in the soluble polysaccharides content. However, no direct correlation was observed between the changes in the agar and soluble polysaccharide content. Therefore, the soluble polysaccharides content cannot be directly used as an indicator of the agar content.

According to researchers in the field, the transcription level of some agar biosynthesis-related genes can indicate the agar content level. Chen verified that the enzyme activities of PGM and the agar content exhibited significant correlations (Pearson correlation coefficient: 0.896) after 7 days for *G. lemaneiformis*, under the nitrogen concentration treatment [33]. Most *G. lemaneiformis* strains in the high agar content group had higher transcription levels of *galt* than all strains in the low agar content group, which suggested that the *galt* gene played a crucial role in agar biosynthesis [34]. Furthermore, Siow demonstrated that the transcription level of the *galt* gene was higher in *Gracilaria changii* but lower in *Gracilaria salicornia*, which correlated with their respective agar content [27]. Following N-limitation, P-limitation, and low-salinity cultivation for 3 weeks, the agar content of *G. lemaneiformis* exhibited some degree of positive correlation with *mpg* expression [20]. The agar content is a quantitative trait controlled by multiple genes. In this research, 11 agar biosynthesis-related genes correspond to 9 enzymes that cover the core pathway of agar biosynthesis from fructose-6-phosphate to the polymerisation of UDP-D-galactose and GDP-L-galactose to generate agar. Based on the transcriptional analysis of 11 genes, the expression of *gpiI*, *gpiII*, *galt*, *mpi*, *mpg*, and *gatII* showed significant upregulation under low salinity (10‰ and 20‰), high salinity (30‰), high temperature (30 °C), and low P (0.5 μM) treatments (Figure 4), which was followed by a significant increase in agar accumulation under these conditions (Figure 2). Furthermore, the comparison results among different varieties and generations of *G. lemaneiformis* showed that the expression of *gpiI*, *gpiII*, *galt*, *mpi*, *mpg*, and *gatII* in the cultivar 981 *G. lemaneiformis* with the highest agar content were significantly upregulated (Figure 5), which verified that the transcription levels of these genes are correlated with agar accumulation. Through the quantitative statistics of data, the changes in the relative transcription levels of *gpiII*, *mpi*, *mpg*, and *gatII* were strongly and positively correlated with RAA, with PCC of 0.925, 0.776, 0.901, and 0.856, respectively (Table 1). 

Glucose-6-phosphate isomerase (GPI) and mannose-6-phosphate isomerase (MPI) are located at the beginning of the agar biosynthesis pathway, reversibly catalysing fructose-6-phosphate into glucose-6-phosphate and mannose-6-phosphate, respectively. Moreover, the GPI and MPI are also two types of enzymes involved in glycolysis, which is an important pathway for sugar disassimilation in organisms [35]. It has been reported that the expression of GPI in plant cells may be affected by environmental conditions. In investigating the effect of temperatures on the pathways of glycolysis and alcoholic fermentation in shoot and root tissues of rice seedlings, *gpi* transcript changed in response to high-temperature stress. Evidently, glycolytic and alcohol fermentation enzymes have a sufficient degree of flexibility to adjust to increased energy demand and supply of intermediates for acclimatising to stress conditions [36]. Real-time quantitative RT-PCR demonstrated that the expression level of the *gpi* gene from *Dunaliella salina* (DsGPI) was induced by 3.5 M NaCl with 14-fold higher levels than that by 1.5 M NaCl (*p* < 0.01) [37]. Under the stress of this study, the transcription levels of the enzyme increased significantly, on the one hand, by catalysing the synthesis of agar to enhance cell toughness, and, on the other hand, by enhancing sugar metabolism to produce more energy to resist stress. The transcript level of the *mpg*, which expresses GDP-mannose pyrophosphorylase that catalyzes the formation of GDP-D-mannose, was highly correlated with agar content. According to the report, repression of GDP-mannose pyrophosphorylase in yeast, which has the same conservative structural domain as mannose-1-phosphate guanylyltransferase of *G. lemaneiformis*, led to phenotypes, such as cell lysis and a defective cell wall [38]. Moreover, GDP-mannose pyrophosphorylase plays an essential role in cell wall integrity, morphogenesis, and viability [39]. It can be seen that *mpg* plays an important role in agar biosynthesis and cell wall assembly. Glycosyltransferase II (GATII) belongs to the glycosyltransferase (GT) family to facilitate fast assembly of nucleotide sugars into polysaccharide chains in the Golgi apparatus [40,41]. These enzymes from the GT family are involved in cell wall metabolism but with a surprisingly low redundancy in *Chondrus crispus* [42]. Chen confirmed that the expression level of GT7-2 was positively correlated with agar accumulation in *G. lemaneiformis* [29]. Overall, the four enzymes were more active under stress conditions and the cultivar 981 with a higher agar content, which could be used as the indicator for agar accumulation in *G. lemaneiformis* (Figure 7).

## 4. Materials and Methods

### 4.1. Algae and Homogenisation Cultivation

The wild-type *Gracilariopsis lemaneiformis* (Bory) E.Y. Dawson, Acleto & Foldvik 1964 (Rhodphyta), was collected from the intertidal zone of Fushan Bay (36.0° N, 120.3° E) in Qingdao, Shandong Province, China. The cultivar 981 *G. lemaneiformis*, being farmed industrially, was obtained from the seed storage of our laboratory.

In the laboratory, the algae were divided into several parts, which were almost identical in terms of length, branch number, and diameter. The algae were cultivated in the transparent tank at an irradiance of 40 ± 3 µmol photons m^−2^·s^−1^ and 12 h:12 h light/dark cycle at 20 °C [33]. The density of cultivation was that every 5 g of algae was submerged in 1 L filtered natural seawater (salinity of 30‰). The seawater was refreshed every 2 days. 

### 4.2. Cultivation under Four Treatments

After homogenisation cultivation for 4 days, the algae with good growth status were cultured under different salinity, temperature, nitrogen concentrations, and phosphorus concentrations, respectively.

The algae were equally divided into four groups (four levels), with each group containing five biological replicates. After continuing to be homogenised for 3 days, 4 groups of algae were cultivated in seawater with different treatments for 15 days, respectively. The treatment schemes of different salinities, temperatures, nitrogen (N), and phosphorus (P) concentration treatments were similar, and the detailed setting was shown in Table 2. 

The various salinities were achieved by increasing the content of ultrapure water and salt in filtered natural seawater. The culture medium for algae treated under different N and P concentrations was artificial seawater (Table 3). For N concentration, 6 μM is close to the N concentration in Qingdao offshore water, 30 μM is the minimum threshold N concentration for near eutrophic seawater, and 300 μM and 900 μM are close to the N concentrations of the artificial seawater and f/2 culture medium, respectively [43,44]. For P concentration, 0.5 μM is the P concentration close to Qingdao offshore water, 1.5 μM is the minimum threshold P concentration for near eutrophic seawater, and 30 μΜ is the phosphorus concentration of the artificial seawater for culture medium. The detailed setting for N and P treatments is shown in Table 4.

The culture medium was refreshed every 2 days and the cultivation density was the same as that of assimilation cultivation.

### 4.3. Determination of Relative Growth Rate

The change of biomass (Fresh Weight, FW) in algae was measured at 0, 3, 6, 9, 12, and 15 days and each group contained 5 biological replicates. The relative growth rate (RGR) was determined in order to reflect the physiological performance of algae. The detailed formula for calculating the RGR was as follows [45]:RGR (% d^−1^) = [(lnW − lnW_0_)/t] × 100
where W and W_0_ are the final and initial FW of each individual, respectively, and t is the time of cultivation in a day.

### 4.4. Determination of Agar Content

The algae were evenly divided into two parts from the base, half of which was taken for extracting agar at 0 days and the other half of which was used for extracting agar after 15 days of treatments, and each group was set to 5 biological replicates. Approximately 5 g sample (about 0.7 g dry weight) was used to extract agar applying the alkali treatment method with slight modification [20]. The procedure was as follows:

The sample, the crystal salt on the surface of which was washed away, was dried to a constant weight at 65 °C and the dry weight was determined. Sodium hydroxide solution (2.5% *w*/*v*) was added to the dry sample (4 mL sodium hydroxide solution per 0.1 g dry weight) and heated in a water bath at 85 °C for 2 h. The sample was washed with ultrapure water until the pH of the washing solution dropped to around 6.5. Then, ultrapure water was added to the sample (6 mL of ultrapure water per 0.1 g dry weight) and treated at 120 °C for 40 min. The mixture was squeezed and filtered, and the filtrate was collected. The filter residue was recycled and carried the secondary cooking (the volume of the ultrapure water added was 10 mL less than that of the first), squeeze, and filter process. The filtrate cooled to room temperature was frozen overnight at −20 °C. The frozen gel was thawed, washed with distilled water, and dried to constant weight at 65 °C. The detailed formula for calculating the agar content was as follows:Agar content (%) = dry weight of agar/dry weight of sample × 100(1)
Accumulation of agar = agar content at 15 days − agar content at 0 days(2)
Relative accumulation of agar (RAA) = accumulation of agar for experiment group − accumulation of agar for control group(3)
RAA of 981 = agar content at the 981 − agar content at the WT-D(4)

### 4.5. Measurement of Soluble Polysaccharides Content

The algae excluding the tip and base was selected to extract soluble polysaccharides at 0, 3, 6, 9, and 15 days. According to Liu’s method with slight modification, about 0.1 g of the powdered sample (FW) obtained by the liquid nitrogen grinding method was mixed with 7 mL of ultrapure water and heated in a water bath at 80 °C for 2 h, and shaking every 30 min [32]. After centrifugation at 3381 g for 10 min, the supernatant was collected for measurement of soluble polysaccharides content.

The content of soluble polysaccharides was determined by the phenol–sulfuric acid method, with glucose as the standard and ultrapure water as the blank [46]. An amount of 200 μL of supernatant, 800 μL of ultrapure water, and 1 mL of 6% phenol were mixed, followed by the quick addition of 5 mL of concentrated sulfuric acid and then shaking it well. After cooling down, the mixture was heated in a water bath at 40 °C for 20 min. The final solution was cooled down to room temperature and measured using a 721 Visible Spectrophotometer (YOKE instruments, Shanghai) at 490 nm (the size of the cuvette: 10 × 10 × 45 mm). The exact content of soluble polysaccharides was calculated using the standard curve method (the glucose concentration range for the standard curve: 0–0.18 mg/mL). Using the formula derived from the standard curve, the soluble polysaccharide concentration of the sample was obtained. The value obtained by multiplying this concentration value by the dilutions (5) and the volume (7) of water was the weight of the soluble polysaccharide of the sample (mg). The detailed formula for calculating the soluble polysaccharide (SP) content was as follows:SP content (mg·g^−1^) = weight of soluble polysaccharides (mg)/weight of powdered sample (g)

### 4.6. RNA Extraction and cDNA Synthesis

Total RNA was extracted from the tip of thalli at 0, 1, 2, 3, 6, and 9 days according to the instruction manual of Plant RNA Kit (Omega, Washington, DC, USA), and the principle of sampling was cross sampling between multiple algae strains. The quality of RNA samples was examined by agarose gel electrophoresis and NanoDrop (OD_260/280_ and OD_260/230_). The cDNA was synthesised according to the instruction manual of Prime Script™ RT Master Mix (Perfect Real Time) (Takara Bio, Osaka, Japan) and stored at −40 °C.

### 4.7. Analysis of Genes Transcription Levels

Eleven genes involved in the agar biosynthesis, including glucose-6-phosphate isomerase I (*gpiI*) [47], glucose-6-phosphate isomerase II (*gpiII*) [47], Phosphoglucomutase (*pgm*), UDP-glucose pyrophosphorylase (*ugp*), galactose-1-phosphate uridylyltransferase (*galt*) [34,47], mannose-6-phosphate isomerase (*mpi*), phosphomannomutase (*pmm*), mannose-1-phosphate guanylyltransferase (*mpg*), GDP-mannose-3,5′-epimerase (*gme*), galactosyltransferase I (*gatI*), and galactosyltransferase II (*gatII*), were analysed. The actin gene was used as the housekeeping gene [48,49]. Complete sequence information for all genes was presented in the Appendix A (Sequences). All primers (Table 5) were designed using the Primer 5.0 software (Primer, Toronto, ON, Canada).

The qRT-PCR (TB Green Premix Ex Taq^TM^ II Kit, Takara) was carried out by LightCycler 480 System (Roche Diagnostics, Basel, Switzerland). The program used for all samples was as follows: 95 °C for 20 s, followed by 40 cycles of 95 °C for 15 s, 55 °C for 15 s, and 72 °C for 20 s, and reading the fluorescence signal, followed by 1 cycle of 95 °C for 15 s, 60 °C for 60 s, and 95 °C for 15 s [50]. Each reaction was set to three replicates. The transcription of each gene in the control group was used as a calibrator to determine the relative change in the experimental group. The relative transcription level was calculated by the 2^−∆∆Ct^ (RQ) method [51]. The relevant formulas were as follows:∆Ct = Ct (target gene) − Ct (house-keeping gene)(5)
∆∆Ct = ∆Ct (experiment gruop) − ∆Ct (control group)(6)

The RQ values (the relative transcription level) of each target gene from the experimental group at 0, 1, 2, 3, 6, and 9 days were standardised in the form of LogRQ. The change in the relative transcription level, which refers to the C-value in the relative expression of genes between the post-treatment (1, 2, 3, 6, and 9 days) and pre-treatment (0 days) periods, was visualised using TBtools [24]. The detailed formula for calculating the C-value was as follows:C-value = LogRQ (1, 2, 3, 6, and 9 days) − LogRQ (0 days)

### 4.8. Statistical Analysis

Statistical analyses of the datasets for the agar content, soluble polysaccharides content, and growth rate were performed using SPSS 25.0 software (IBM Corp, Armonk, NY, USA). The normal distribution and the homogeneity of different datasets were confirmed by a Shapiro–Wilk test (*p* > 0.05) and Levene’s test (*p* > 0.05), respectively. The one-way analysis of variance (ANOVA) was performed to determine the differences between the different datasets using Bonferroin’s post hoc test and Tukey’s post hoc test (Tukey HSD) [33]. If some datasets did not conform to the normal distribution, the Kruskal–Wallis test was used to analyse them [52]. Pearson correlation analysis was used to detect the correlation between two variables. The significance level was set at *p* < 0.05. 

## 5. Conclusions

Regarding the current fragmentary understanding of the agar biosynthesis pathway, we comprehensively discussed the mutual relation between the agar content and the relative growth rate, soluble polysaccharides content, and expression level of multiple genes involved in the agar biosynthesis pathway of *Gracilariopsis lemaneiformis* in this study. Under different treatments, the growth and soluble polysaccharides content of algae showed more significant change compared to the agar content, which proved the relative stability of agar. Moreover, there is no significant correlation between the agar content, as colloidal polysaccharides, and the soluble polysaccharides content. However, the expression level of *gpiII*, *mpi*, *mpg*, and *gatII* showed a high correlation with the agar content.

Improving agar content is one of the most important aims of cultivated Gracilariaceae. From the commercial point of view, it is optimal to obtain algae with high growth rates, high agar yield, and superior gelling properties. Improving the agar content as the main economic trait of *G*. *lemaneiformis* is the goal that we are striving to achieve. Molecular indicators provide us with excellent markers in the screening of breeding materials, which will be helpful for promoting the development of the *G. lemaneiformis* cultivation industry. 

## Figures and Tables

**Figure 1 ijms-25-04606-f001:**
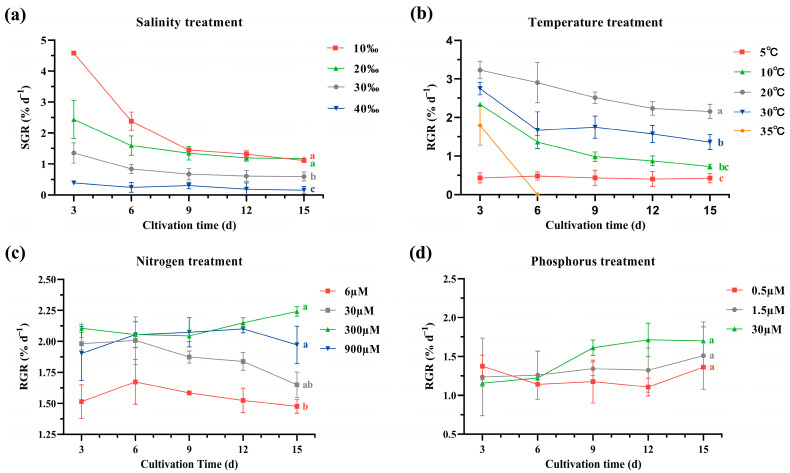
The RGR of *G. lemaneiformis* under different treatments: (**a**) salinity, (**b**) temperature, (**c**) nitrogen, and (**d**) phosphorus. The datasets of salinity and nitrogen treatments did not conform to the normal distribution. The differences between the two datasets were detected separately using Kruskal–Wallis test followed by Bonferroin’s post hoc test. The differences in letters on the right in the subfigure represent significant differences between groups.

**Figure 2 ijms-25-04606-f002:**
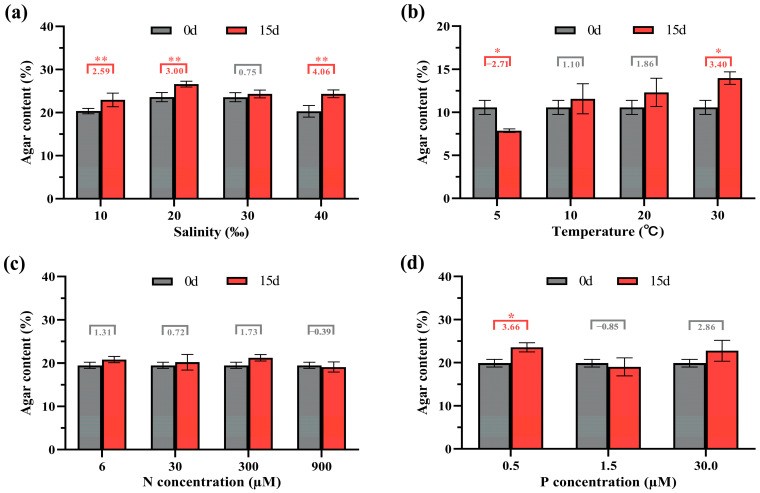
The agar content of *G. lemaneiformis* under different treatments: (**a**) salinity, (**b**) temperature, (**c**) nitrogen, and (**d**) phosphorus. * (*p* < 0.05) and ** (*p* < 0.01) represent the significant differences and highly significant differences, respectively.

**Figure 3 ijms-25-04606-f003:**
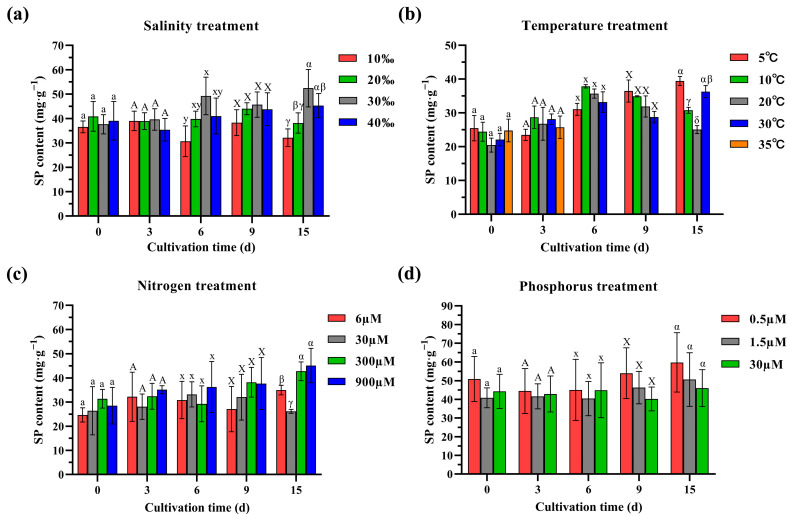
The soluble polysaccharides (SP) content of *G. lemaneiformis* under different treatments: (**a**) salinity, (**b**) temperature, (**c**) nitrogen, and (**d**) phosphorus. The datasets were analysed, with different letters (*p* < 0.05) considering statistical differences between different groups (levels) at the same time.

**Figure 4 ijms-25-04606-f004:**
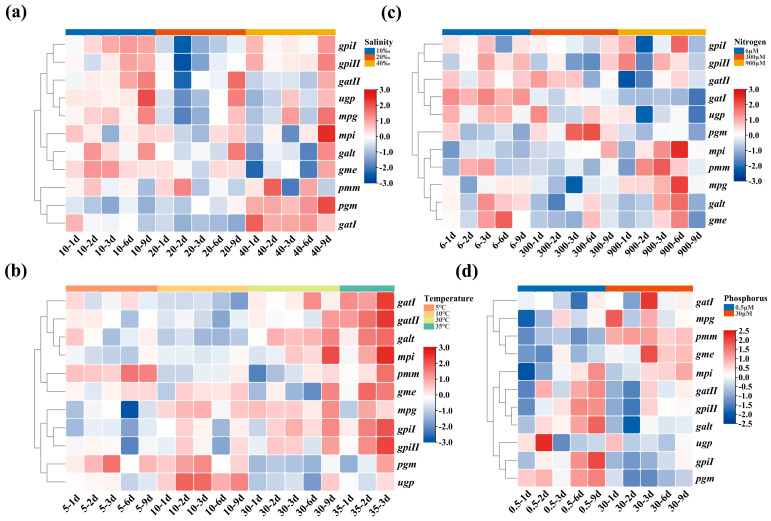
The expression levels of the 11 agar biosynthesis genes of *G. lemaneiformis* under salinity treatments. (**a**) salinity, (**b**) temperature, (**c**) nitrogen, and (**d**) phosphorus. The C-value was visualised and data with a similar trend were clustered based on Euclidean distance using TBtools. TBtools was reprinted with permission from Reference [24]. 2020, Chengjie Chen.

**Figure 5 ijms-25-04606-f005:**
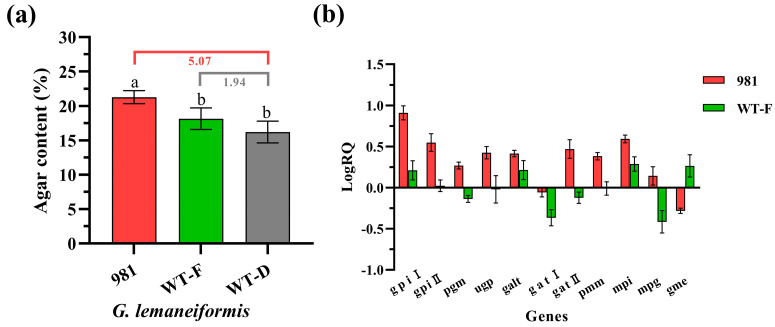
The agar content and 11 genes transcription level in different varieties and generations of *G. lemaneiformis* (981: the cultivar 981; WT-F: the wild-type female gametophyte; WT-D: the wild-type diploid. (**a**) Agar content, the difference of the lowercase letters presents the statistical differences between the agar content of different *G. lemaneiformis*; (**b**) transcription level of 11 genes involved in agar biosynthesis. The transcription of each gene in the WT-D was used as a calibrator to determine the relative change in the 981 and WT-F. ∆∆Ct = ∆Ct (the 981 or WT-F) − ∆Ct (the WT-D). The relative transcription level was calculated by the 2^−∆∆Ct^ (RQ) method. The RQ value was standardised in the form of LogRQ.

**Figure 6 ijms-25-04606-f006:**
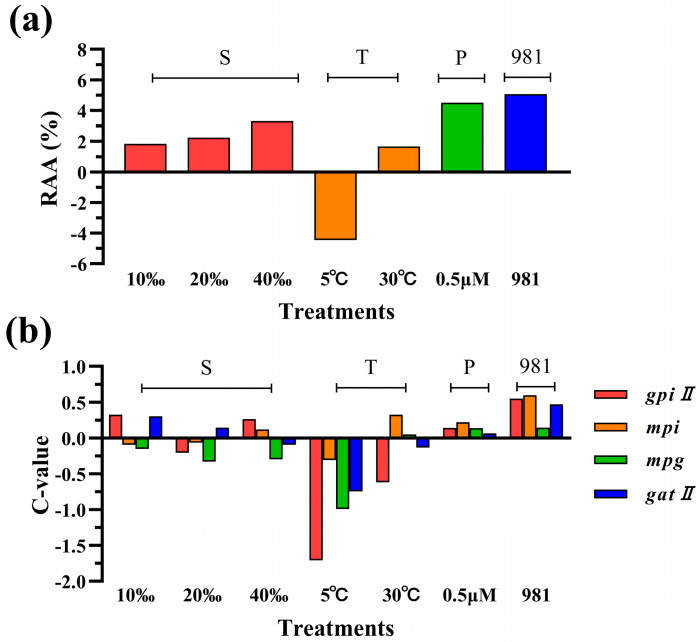
The comparative analysis of the significant changes in agar content and gene transcription levels. S: salinity, T: temperature, P: phosphorus, 981: the cultivar 981. (**a**) The relative accumulation of agar (RAA). (**b**) C-value.

**Figure 7 ijms-25-04606-f007:**
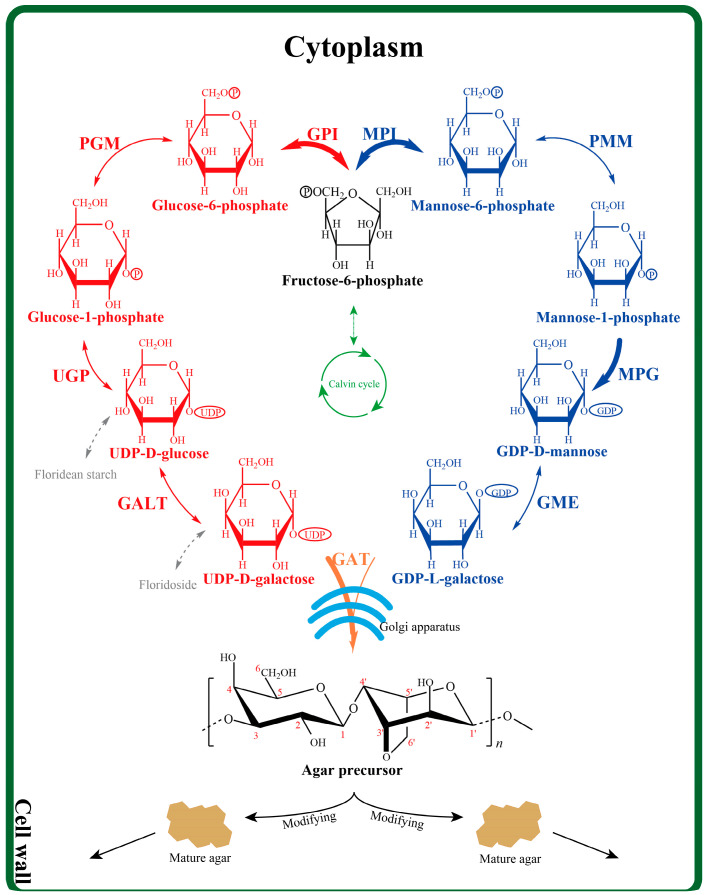
The proposed agar biosynthesis pathway in red algae, which was adapted with permission from Reference [21]. 2017, Wei-Kang Lee. The pathway starts with the polymerisation of UDP-D-galactose and GDP-L-galactose, which are synthesised from fructose-6-phosphate derived from Calvin cycle, or by the degradation of floridean starch and floridoside. GPI: glucose-6-phosphate isomerase (*gpiI* and *gpiII*); PGM: phosphoglucomutase (*pgm*); UGP: UDP-glucose pyrophosphorylase (*ugp*); GALT: galactose-1-phosphate uridylyltransferase (*galt*); MPI: mannose-6-phosphate isomerase (*mpi*); PMM: phosphomannomutase (*pmm*); MPG: mannose-1-phosphate guanylyltransferase (*mpg*); GME: GDP-mannose-3,5′-epimerase (*gme*); GAT: glycosyltransferase (*gatI* and *gatII*). The numbers contained in the agar precursors represent the carbon atom order of the galactose units. For example, 1 and 1’ refer to the first carbon atom of D-galactose and L-galactose, respectively. The bold arrow indicates the metabolic process most relevant to agar content obtained in this study.

**Table 1 ijms-25-04606-t001:** The PCC between the C-value of the 11 genes and RAA.

Genes	*gpiI*	*gpiII*	*pgm*	*ugp*	*galt*	*mpi*	*pmm*	*mpg*	*gme*	*gatI*	*gatII*
PCC	0.503	0.925 *	−0.278	−0.023	0.742	0.776 *	−0.532	0.901 *	−0.636	0.209	0.856 *

* (*p* < 0.05) represents the significant correlation.

**Table 2 ijms-25-04606-t002:** The detailed setting under four treatments.

Sample Collection Time	Treatments	Groups (Levels)				
13 August 2022	Salinity (‰)	10	20	30 (control)	40	-
3 September 2022	Temperature (°C)	5	10	20 (control)	30	35
12 October 2022	N (µM)	6	30 (control)	300	900	-
11 November 2022	P (µM)	0.5	1.5 (control)	30	-	-

**Table 3 ijms-25-04606-t003:** The artificial seawater formula.

Component	Concentration (g/L)
NaCl	24.470
Na_2_SO_4_	3.9170
KCl	0.6640
KBr	0.0960
SrCl_2_·6H_2_O	0.0402
MgCl·6H_2_O	4.9810
CaCl_2_	0.9480
NaHCO_3_	0.1920
NaF	0.0039
NaH_2_PO_4_·2H_2_O	X
H_3_BO_3_	0.0260
NaNO_3_	Y
Na_2_SiO_3_·9H_2_O	0.0230

The artificial seawater formula was slightly modified based on the report by adapted with permission from Reference [43]. 2001, Berges.

**Table 4 ijms-25-04606-t004:** The detailed setting for N and P treatments.

Treatments	X, g/L (µM)	Y, g/L (µM)
Various N Concentrations	4.68 × 10^−3^ (30)	5.10 × 10^−5^ (6)
		2.55 × 10^−3^ (30)
		2.55 × 10^−2^ (300)
		7.65 × 10^−2^ (900)
Various P Concentrations	7.80 × 10^−5^ (0.5)	2.55 × 10^−2^ (300)
	2.35 × 10^−4^ (1.5)	
	4.68 × 10^−3^ (30)	

**Table 5 ijms-25-04606-t005:** Primers information in this study.

Genes Name	Forward Primers (5′–3′)	Reverse Primer (5′–3′)	PCR Products (bp)
*gpiI*	TGCTCCAACTTGCTGCCGA	GCTCGCAGGGCGGTATGAA	193
*gpiII*	ACCACCGCCGAAACCATGC	CCAGTCCCAGAATCCAAACACG	165
*pgm*	TCCTTCTGATTCTGTCGCTGTC	ATTTCCAACCTGTAGGCACTTC	167
*ugp*	GACCTTATCGTCCAGCAAATCG	CCTTGACAATCCTCGGGTAGCG	181
*galt*	CGTTCCTTTGCTTCCATTTCG	TCTGGGCATCTGAGTTCGTTC	127
*mpi*	CTCCGCCGACTCCGTTCAAGA	TACGCCGCAAAGCAGCCCACAT	138
*pmm*	CGTTCTGGGATTACGGTCTC	GCATAGTTTCTTCCGGGTTC	219
*mpg*	CCTTGATTCTCGTCGGTGGTTA	GCATCTTCTCGGGCTGGTAGTT	174
*gme*	TGGGTAAGATGGTGCTCGGATTT	TCAAAAGTCCTCCTGAGCCCGT	163
*gatI*	GCTGGATTTGTAATGTTGATGTTGC	TGGATAAGCCTCGTGCGTTC	211
*gatII*	CCTTGAGCTGGTTGTGCAATGT	CATGAGCATCCAGTATCCCG	233
*act*	CTACTCGTTTACCACTTCTGCTGA	TTCCATTCCGACCAACTCTG	220

Additional information about the primers’ characteristics was shown in the Appendix A.

## Data Availability

Data will be made available on request.

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
