# Peer review of "Identification of Indicator Genes for Agar Accumulation in Gracilariopsis lemaneiformis (Rhodophyta)"

_ijms, 2024, doi:10.3390/ijms25094606_

Round 1
Reviewer 1 Report (Previous Reviewer 3)
Comments and Suggestions for Authors
Manuscript ijms-2955046 aims to investigate the important biotechnological process of agar production. The information presented by the authors is of considerable interest. However, some parts of the manuscript require clarification.
1. Lines 105-108: “Although the algal RGR was the highest in 10‰ salinity (2.17%/day) compared with other salinity levels, the RGR fluctuated markedly in 10‰ salinity (Standard deviation: 1.36) and exhibited a gradually decreasing trend (Figure 1 (a)).” In Figure 1(a), the RGR at 10% and 20% is not a significant difference (both are indicated by the letter “a”). The validity of the “decreasing trend” is not presented.
2. Lines 108-109: “Compared with that in 30‰ salinity (0.81%/day), the algae had a higher RGR in 20‰ (1.55%/day) salinity and a lower RGR in 40‰ salinity (0.24%/ day).” In Figure 1(a), the RGR at 30% and 40% are not a significant difference (both are labeled “b”).
3. Lines 114-119: “Compared with the N concentration of 6 μM (1.55%/day), the 300 μM (2.19%/day) and 900 μM (2.02%/day) N concentration significantly increased the algal RGR, but the RGR exhibited no further increase at the 900 μM N concentration (Figure 1 (c)).” Why is indicated "no further increase at the 900 μM N concentration"? The values for 300 μM (2.19%/day) and 900 μM (2.02%/day) N concentration have already been given previously.
4. Lines 121-122: “The nitrogen concentration of 300 μM is sufficient for the growth of algal strains.” In Figure 1 (c), RGR at 30 μM is not significantly less than RGR at 300 μM. Therefore, 30 μM is sufficient for the growth of algal strains.
5. Lines 122-125: “When the phosphorus concentration increased, the algal growth rate increased (Figure 1 (d)). However, significant difference was not noted among the effect of 0.5 μM (1.23%/day), 1.5 μM (1.34%/day), and 30 μΜ (1.49%/day) P concentrations on algal RGR. 0.5 μM P concentration can weakly support G. lemaneiformis growth.” Why do the authors use "weakly" here? After all, the results do not show that any phosphorus concentration increases RGR.
6. Lines 126-127: “In short, adding a certain concentration of nitrogen and phosphorus into seawater is beneficial for algal growth in the laboratory (Figure 1 (c and d)).” Indicate here the exact concentrations of nitrogen and phosphorus that are beneficial for algal growth in the laboratory.
7. In Figure 2 (a), the axis labels must be in English.
8. Table 5. Replace “N” with “Various Nitrogen Concentrations." Replace “P” with “Various Phosphorus Concentrations.”
9. Glucose-6-phosphate isomerase (GPI) is an enzyme also involved in glycolysis and gluconeogenesis. Activation of the gpi gene can depend on various environmental changes and increase the rate of different metabolic pathways. This is not discussed in the manuscript.
Author Response
Thanks for your suggestions.

Reviewer 2 Report (New Reviewer)
Comments and Suggestions for Authors
This paper examined red algae, which is known for its agar production, observing changes in its growth, agar yield, polysaccharide levels, and expression of genes related to polysaccharide synthesis under altered growing conditions.
The stated aim of this journal, as per its website, is to serve "as an advanced forum for biochemistry, molecular and cell biology, molecular biophysics, molecular medicine, and all aspects of molecular research in chemistry." Based on this, this particular study does not seem to be a good fit for the journal.
Even if it did fit, I understand that the alga is of a practical strain, however, the data seems to be quite basic and preliminary to warrant a single article and is significantly lacking in novelty.
Author Response
Thanks for your suggestions.

Round 2
Reviewer 2 Report (New Reviewer)
Comments and Suggestions for Authors Thank you for your explanation about the originality of the paper. I now fully understand the significance of this paper. As you outlined in your explanation, please clarify the purpose and significance of this paper. Additionally, I request a revision on one point: the way the data is presented in Table 2. It might be difficult for readers to understand, despite its importance. For example, the complete data should be presented, not just the numbers after the decimal point. Also, is it possible to calculate the standard deviation (SD) or other measures of error for the PCC? If so, this should be included in the data. Moreover, the meaning of "Sig." is unclear. If it represents the significance (p-value), perhaps it's enough to denote P<0.05 just with an asterisk (*) and there might be no need to show the number, particularly if the SD can be displayed. In relation to Table 2, I was unable to locate Table 1.Author Response
Thanks for the suggestion.

This manuscript is a resubmission of an earlier submission. The following is a list of the peer review reports and author responses from that submission.
Round 1
Reviewer 1 Report
Comments and Suggestions for Authors
Overall, the manuscript is well-founded and the experimental part is well designed.
Regarding the formal part of the writing, I recommend that the numerous mistakes be corrected, as there is no adequate space between the quotations and the respective sentences.
Here are some example situations:
Corrections needed (some examples):
line 30 - charides [1,2,3,4]. The cell wall polysaccharides plays vital roles for algae in the adaptation
line 98 - tuates greatly in 10‰ salinity (Standard deviation: 1.36) and showed a gradually decreas-
line 99 - ing trend (Figure 1 (a)). Compared with 30‰ salinity (0.81%/d), the algae had higher
line 138 - 300 μM and 900 μM. Compared with the initial agar content, the agar content of thalli cul-
and so on ...
The manuscript entitled "Title Identification of Key genes for Agar Biosynthesis in Gracilariopsis lemaneiformis (Rhodophyta) under Different Treatments" addresses a very relevant and appropriate topic for this journal.
The manuscript is well structured and, overall, well-researched. In terms of writing, authors will have to create spaces between the units and their respective values.
Bibliographic references between [] must be separated by a space, from the end of each sentence.
Corrections needed:
line 45 - vascular plants and red seaweeds [14]. The proposed pathway starts from fructose-6-phosphate,
line 148 - Figure 2. Agar content of G. lemaneiformis under different treatments. (a) salinity, (b) temperature, (Note: "G. lemaneiformis" in italics)
line 176 - Figure 3. Soluble polysaccharides (SS) content of G. lemaneiformis under different treatments. (a) (Note: "G. lemaneiformis" in italics)
line 220 - Figure 4. Expression levels of the 11 genes of G. lemaneiformis under salinity treatments. (a) Salinity, (Note: "G. lemaneiformis" in italics)
line 262 - Previous studies have shown that in other agarophytes ... (Note: The authors should add more examples of other agarophyte species, both of the genus Gracilaria and other genera (for example, Pterocladiella, Gacilariopsis, Gelidium, etc.) to this discussion section))
line 374 - Remove this paragraph:
"The Materials and Methods should be described with sufficient details to allow 375
others to replicate and build on the published results. Please note that the publication of 376
your manuscript implicates that you must make all materials, data, computer code, and 377
protocols associated with the publication available to readers. Please disclose at the 378
submission stage any restrictions on the availability of materials or information. New 379
methods and protocols should be described in detail while well-established methods can 380
be briefly described and appropriately cited. 381
Research manuscripts reporting large datasets that are deposited in a publicly 382
available database should specify where the data have been deposited and provide the 383
relevant accession numbers. If the accession numbers have not yet been obtained at the 384
time of submission, please state that they will be provided during review. They must be 385
provided prior to publication. 386
Interventionary studies involving animals or humans, and other studies that require 387
ethical approval, must list the authority that provided approval and the corresponding 388
ethical approval code."
line 391 - Wild Gracilariopsis lemaneiformis (Bory) E.Y. Dawson, Acleto & Foldvik 1964 (Rhodphyta), was collected from the intertidal zone of Fushan Bay
Minor revision

Comments on the Quality of English LanguageMinor editing of English language required
Author Response
Thank you for your suggestions. The corresponding information has been added to the following document.

Reviewer 2 Report
Comments and Suggestions for Authors
The paper of Li et al. entitled “Identification of Key genes for Agar Biosynthesis in Gracilariopsis lemaneiformis (Rhodophyta) under Different Treatments” fails to convince that the findings can be used as an indicator of agar accumulation in this red alga. First of all, looking at gene expression is not the best tool to understand what is going on during different treatments, that is, salinity, temperature, and nutrients. Transcriptomics, in general, is a poor indicator of complex biochemical processes. Proteomics and metabolomics provide a much better understanding of these processes.
The pattern of expression of the 11 genes is very different when looking at various salinity, temperature, nitrogen, and phosphorous conditions. The idea of a relevant correlation is not clear from the data. The composite analyses from Table 1 and Figure 5 are not convincing. If a strong positive correlation was found between gpi, mpi galt and mpg on day 9, what about the other days? Quite interestingly, day 9 is also the last time point! Theoretically, the experimental design was not OK because it is essential to know if this trend is maintained or not after day 9!
This is one of the multiple issues with this paper. The usage of the English language is rather poor, that is, there are many instances in which the text has to be revised, misspelling can be found everywhere in the text, including in figures (for example, Figure 3 - Cltivation time instead of cultivation time); the text and especially references were not well formatted.
To conclude, the topic was not addressed using the correct approach, that is, transcriptomics was used instead of proteomics and metabolomics. The experimental design is not OK, as data collection stopped on day 9, the only one in which some relevant data is suggested. Because of the problems with the method used, and with the experimental design, the conclusions of this work are not sustainable.
Comments on the Quality of English LanguageEnglish language usage is poor, the MS needs significant revisions.
Author Response
Response to Reviewer 1 Comments
Point 1: Regarding the formal part of the writing, the numerous mistakes be corrected, as there is no adequate space between the quotations and the respective sentences, between the units and their respective values, and so on.
Response 1: Thanks for your suggestion.
We have made change for these mistakes.
Point 2: line 252 - Previous studies have shown that in other agarophytes ... (Note: The authors should add more examples of other agarophyte species, both of the genus Gracilaria and other genera (for example, Pterocladiella, Gacilariopsis, Gelidium, etc.) to this discussion section))
Response 2: Thanks for your suggestion.
These additions have been presented (line 253-262 in manuscript). The detailed information is as follows:
The agar yield of agarophytes is related to water temperature [38]. The agar content of Gracilaria bursa-pastoris was positively correlated with temperature (r = 0.94; P < 0.01) [4]. Friedlander found that the agar yield of Gracilaria conferta cultured in tanks was positively correlated with water temperature [41]. In Chen's study, the agar content of G. lemaneiformis cultured under heat stress (28 °C) for 11 days increased compared with that under the control condition (23 °C), but this difference was non-significant [37]. For Gelidium coulteri (with an optimal growth temperature of 25 °C–27 °C), within the temperature range of 5 °C–31 °C, agar accumulation in the algae gradually increased as the temperature increased [47]. However, within the temperature range of 10 °C–30 °C, the agar content of Gracilaria sordida decreased with an increase in temperature [42].
Point 3: line 374 - Remove this paragraph:
Response 3: Thanks for your suggestion.
We have removed this paragraph.
Point 4: line 361 - Wild Gracilariopsis lemaneiformis (Bory) E.Y. Dawson, Acleto & Foldvik 1964 (Rhodphyta), was collected from the intertidal zone of Fushan Bay.
Response 4: Thanks for your suggestion.
This important information has been added(line 361 in manuscript).

Reviewer 3 Report
Comments and Suggestions for Authors
The manuscript ijms-2870801 describes the results of the analysis of the activity of genes for the biosynthesis of red algae agar under various environmental conditions. The authors presented a manuscript of poor quality, which can’t be published in this form.
Major remarks:
1. Authors should carefully read the manuscript and remove errors, unnecessary text fragments, Chinese characters and other inaccuracies from it so that the manuscript meets the requirements of the journal.
2. The rationale for the choice of experimental conditions and conclusions should be formulated more clearly. Avoid drawing conclusions that are not based on the results obtained. For example, line 15: “The agar content was opposite to the relative growth rate of the algae.""—based on the results of which experiments was this conclusion made? Add to the manuscript the results of the final agar accumulation in algae, taking into account the increase in weight over 15 days of cultivation and the content of agar in the thalli.
3. The presentation of statistical analysis results requires improvement. For example, in Figure 1, the differences between the options are not shown. At the same time, the text discusses significant differences between the groups.
4. The primers created by the authors were used in the work. Give the results of electrophoresis on an agarose gel (or other experiment) showing the formation of PCR products when using primers with the specified program (lines 482-485). Content GC (%) in primers is very different, so I have doubts that the formation of PCR products for different primers will be equally effective when using a single program. Provide additional information about the primers' characteristics in the supplementary material.
Minor remarks:
Line 21: “on 9d” change “on 9 day".
Line 25: “agar synthesis related genes” as a keyword? "agar biosynthesis genes” is used more often in the manuscript (3 times).
Lines 58-62: there is a contradiction between” the alga is grown under altered salinity... may reduce the agar production” and “the agar content of Gracilariopsis lemaneiformis was much increased under hypertonic, hypotonic,... cultures". At what salinity can decrease the agar production?
Line 67: “By 2015, The...”
Lines 77–83: specify the links for the information provided.
Line 96 et seq.: What does "SGR" mean? Explain at the first mention.
Lines 105–120: explanations of nitrogen and phosphorus concentrations are a repeat of the section Materials and methods (lines 404–412).
Lines 114–115: The manuscript demonstrates the growth of algae at concentrations from 6 µM up to 900 µM. What does “A nitrogen concentration of 300 µM is sufficient for the growth of algae strains." mean?
Lines 116–122: There is a contradiction between “there was no significant difference in the SGR of algae among phosphorus concentrations...” and “it was beneficial to add a certain concentration of... phosphorus into seawater for the growth of algae in the laboratory”.
Figure 1: What do the bars in the picture show?
Figure 2: What do the bars show in the picture? There are no letters in the drawing (line 149: “different letters”). How do the authors explain the strong differences in agar content (%) on 0 days in different experiments, from 10% (Fig. 2b) to 22% (Fig. 2a)? And within each experiment (about 5%)?
Figure 3: Why are “Soluble polysaccharides” labeled "SS" and not "SP"? What do the bars show in the picture? Use different letters for groups from different times (a, b, c; A, B, C; α, β, γ; x, y, z; X, Y, Z; or others). Otherwise, you need to use the two-way ANOVA.
Lines 180-155: move this fragment to the Materials and Methods section
Line 231: “RAA on 15d” will be correct
Line 270-272: These data do not correspond to the results shown in Figure 2b.
Lines 298-300: algal cells are not plant cells. The reference 43 is incorrect for this part of the manuscript. Provide reference(s) for red algae.
Lines 322-326: Delete this part, or make it clearer and provide references.
Lines 356-359: Delete this part, or make it clearer and provide references.
Figure 6: "UDP-D-mannose" is an incorrect connection name. Change to "GDP-D-mannose".
Lines 371-385: Delete these lines from the manuscript.
Table 3. Use English only.
Table 4: Use the notations “X, g/L" and “Y, g/L".
Line 423: “for every period". Does this mean 3 days?
Line 427: t = 3?
Line 439: What does “at high pressure” mean?
Line 458: specify the model of the spectrophotometer and the size of the cuvette.
Line 459: Specify the glucose concentration range for the standard curve.
Lines 461-462: the formula does not take into account the dilutions used and the volume of water for extraction. What is “weight of soluble polysaccharides(mg)”?
Line 465: “omega bio-tek, America" is incorrect.
Line 466: “strains"?
Line 478: Why is the actin gene naming the target gene? Give the sequence number for the actin gene in the Gracilariopsis lemaneiformis genome.
Line 489: the formula is incorrect. “Ct(target gene) - Ct(target gene)” is zero.
Line 499-500: why was t test conducted, and not one-way ANOVA?
References are not designed according to the rules of the journal.
Author Response
Response to Reviewer 3 Comments
Point 1: Authors should carefully read the manuscript and remove errors, unnecessary text fragments, Chinese characters and other inaccuracies from it so that the manuscript meets the requirements of the journal.
Response 1: Thanks for your suggestion.
We have made detailed corrections for these major errors.
Point 2: The rationale for the choice of experimental conditions and conclusions should be formulated more clearly. Avoid drawing conclusions that are not based on the results obtained. For example, line 15: “The agar content was opposite to the relative growth rate of the algae.""—based on the results of which experiments was this conclusion made? Add to the manuscript the results of the final agar accumulation in algae, taking into account the increase in weight over 15 days of cultivation and the content of agar in the thalli.
Response 2: Thanks for your suggestion.
The statement, “The agar content was opposite to the relative growth rate of the algae." is misrepresented; it should be that the agar accumulation was inversely proportional to the change in relative growth rate of the algae. And the results of the final agar accumulation in algae for 15 days has added to the manuscript (Table 1).
Point 3: The presentation of statistical analysis results requires improvement. For example, in Figure 1, the differences between the options are not shown. At the same time, the text discusses significant differences between the groups.
Response 3: Thanks for your suggestion.
The statistical analysis of the data in this article has been improved and discussed the significant differences between each group (level) of RGR in algae (Figure 1).
Point 4: The primers created by the authors were used in the work. Give the results of electrophoresis on an agarose gel (or other experiment) showing the formation of PCR products when using primers with the specified program (lines 482-485). Content GC (%) in primers is very different, so I have doubts that the formation of PCR products for different primers will be equally effective when using a single program. Provide additional information about the primers' characteristics in the supplementary material.
Response 4: Thanks for your suggestion.
The results of electrophoresis on an agarose gel and melting curves of the amplification products for the primers are shown in Supplementary Material - Figure (Figure S1 and S2).
Figure S1 The results of electrophoresis on an agarose gel of the amplification products for the primers (Tm = 55 °C)
Figure S1. lane M: DNA Marker (bp), lane 1: gpiI, lane 2: gpiII, lane 3: pgm, lane 4: ugp, lane 5: galt, lane 6: gatI, lane 7: gatII, lane 8: mpi, lane 9: pmm, lane 10: mpg, lane 11: gme , lane 12: act.
Figure S2 The melting curves of the amplification products for the primers
Point 5: Lines 59-64: there is a contradiction between” the alga is grown under altered salinity... may reduce the agar production” and “the agar content of Gracilariopsis lemaneiformis was much increased under hypertonic, hypotonic,... cultures". At what salinity can decrease the agar production?
Response 5: Thanks for your suggestion.
According to the studies that have been done, the agar content of algae is elevated under some degree of low or high salinity transient incubation. At what salinity level the agar content decreases, this study has not been reported. Based on the different growth states of the algae, different salinities may have different differences for them.
Point 6: Lines 79–86: specify the links for the information provided.
Response 6: Thanks for your suggestion.
Complete sequence information for all genes is presented in supplementary material - Sequences.
Point 7: What does "SGR" mean?
Response 7: Thanks for your suggestion.
RGR is correct, and this error has been revised in the original article.
Point 8: Lines 105–120: explanations of nitrogen and phosphorus concentrations are a repeat of the section Materials and methods (lines 404–412).
Response 8: Thanks for your suggestion.
Repeat information has been removed.
Point 9: Lines 113–115: The manuscript demonstrates the growth of algae at concentrations from 6 µM up to 900 µM. What does “A nitrogen concentration of 300 µM is sufficient for the growth of algae strains." mean?
Response 9: Thanks for your suggestion.
The RGR of the algae was similar under nitrogen concentration of 300 µM and 900 µM nitrogen conditions. Therefore, under this culture condition, the addition of 300 nitrogen can meet the basic growth requirement of algae.
Point 10: Lines 116–120: There is a contradiction between “there was no significant difference in the SGR of algae among phosphorus concentrations...” and “it was beneficial to add a certain concentration of... phosphorus into seawater for the growth of algae in the laboratory”.
Response 10: Thanks for your suggestion.
When the phosphorus concentration increased, the algal growth rate increased (Figure 1 (d)). Nitrogen and phosphorus are necessary for the growth of algae. When the concentrations of phosphorus added to seawater is greater than 30 µM, the RGR of the algae may increase significantly.
Point 11: Figure 1: What do the bars in the picture show?
Response 11: Thanks for your suggestion.
The bars show the standard deviation of the dataset in biological replicates.
Point 12: Figure 2: What do the bars show in the picture? There are no letters in the drawing (line 149: “different letters”). How do the authors explain the strong differences in agar content (%) on 0 days in different experiments, from 10% (Fig. 2b) to 22% (Fig. 2a)? And within each experiment (about 5%)?
Response 12: Thanks for your suggestion.
The bars show the standard deviation of the dataset in biological replicates.
Regarding the different letters, this is not a carefully written, and We have corrected it.
As a macroalga, there is a large individual variation in the agar content of G. lemaneiformis. In each treatment, the collection time of the materials used was different (Table 3). Therefore, the growth status of the materials was different, leading to large differences in agar content. However, in one experimental treatment, we strictly standardised the consistency of diameter, length and time and space of growth of the experimental materials, and increased the number of replicates to minimise the error as much as possible.
Point 13: Figure 3: Why are “Soluble polysaccharides” labeled "SS" and not "SP"? What do the bars show in the picture? Use different letters for groups from different times (a, b, c; A, B, C; α, β, γ; x, y, z; X, Y, Z; or others). Otherwise, you need to use the two-way ANOVA.
Response 13: Thanks for your suggestion.
The abbreviation for soluble polysaccharides has been changed to “SP”, and different alphabets were used for groups from different times.
Figure 3. Soluble polysaccharides (SP) content of G. lemaneiformis
Point 14: Lines 180-155: move this fragment to the Materials and Methods section.
Response 14: Thanks for your suggestion.
This fragment has been moved to Lines 468-474.
Point 15: Line 231: “RAA on 15d” will be correct.
Response 15: Thanks for your suggestion.
“RAA at 15 days” has been added to line 221.
Point 16: Line 270-272: These data do not correspond to the results shown in Figure 2b.
Response 16: Thanks for your suggestion.
Compared to algae cultured at 20 °C, the accumulation of agar was significantly reduced at 5 °C and also decreased at 10 °C, so we think that the decrease in the accumulation of agar was basically consistent with the changes in the transcription levels of most agar biosynthesis-related genes, which were generally inhibited at a low temperature (line 264-266).
Point 17: algal cells are not plant cells. The reference 43 is incorrect for this part of the manuscript. Provide reference(s) for red algae.
Response 17: Thanks for your suggestion.
Corresponding references has been provided (line 298).
Point 18: Lines 322-326: Delete this part, or make it clearer and provide references.
Lines 356-359: Delete this part, or make it clearer and provide references.
Lines 371-385: Delete these lines from the manuscript.
Response 18: Thanks for your suggestion.
Because of no convincing reference was found, the corresponding parts (lines 322-326 and lines 356-359) have been deleted.
Point 19: Figure 6: "UDP-D-mannose" is an incorrect connection name. Change to "GDP-D-mannose".
Table 4. Use English only.
Table 5: Use the notations “X, g/L" and “Y, g/L".
Response 19: Thanks for your suggestion.
These have been changed in the paper.
Point 20: Line 397: “for every period". Does this mean 3 days? Line 401: t = 3?
Response 20: Thanks for your suggestion.
Line 397: “for every period", this description is wrong, and it has been corrected, which is “Relative growth rate (RGR) was determined in order to reflect physiological performance of algae”.
Line 401: t = 3, 6, 9, 12, or 15.
Point 21: Line 414: What does “at high pressure” mean?
Response 21: Thanks for your suggestion.
Line 414: “at high pressure” means an environment above standard atmospheric pressure.
Point 22: Line 433-434: specify the model of the spectrophotometer and the size of the cuvette.
Response 22: Thanks for your suggestion.
The model of the spectrophotometer and the size of the cuvette have been specified.
Point 23: Line 435-436: Specify the glucose concentration range for the standard curve.
Response 23: Thanks for your suggestion.
The glucose concentration range for the standard curve has been added.
Point 24: Lines 435-439: the formula does not take into account the dilutions used and the volume of water for extraction. What is “weight of soluble polysaccharides(mg)”?
Response 24: Thanks for your suggestion.
Using the formula derived from the standard curve, the soluble polysaccharide concentration of the sample was obtained. The value obtained by multiplying this concentration value by the dilutions (5) and the volume (7) of water is the weight of soluble polysaccharide of the sample (mg).
Point 25: Line 442: “omega bio-tek, America" is incorrect.
Line 443: “strains"?
Response 25: Thanks for your suggestion.
The instructions for the RNA extraction kit used have been corrected.
The “strains" means different samples of biological replicate groups.
Point 26: Line 454: Why is the actin gene naming the target gene? Give the sequence number for the actin gene in the Gracilariopsis lemaneiformis genome.
Response 26: Thanks for your suggestion.
The correct expression for this should be that “the actin is gene naming the house-keeping gene”. The sequence number for the actin gene in the Gracilariopsis lemaneiformis genome has been presented in supplementary material (Sequences).
Point 27: Line 465 : the formula is incorrect. “Ct(target gene) - Ct(target gene)” is zero.
Response 27: Thanks for your suggestion.
The formula should is “∆Ct = Ct (target gene) – Ct (house-keeping gene)”, which has been corrected.
Point 28: Line 484-485: why was t test conducted, and not one-way ANOVA?
Response 28: Thanks for your suggestion.
For agar extraction, our sampling principle is that “The algae were evenly divided into two parts from the base, half of which was taken for extracting agar at 0 days and the other half of which was used for extracting agar after 15 days of treatments”. The analysis of the dataset for agar content is a comparison before and after treatment of the same sample. Therefore, the agar content data was analyzed using Paired Samples t-test.
Point 29: References are not designed according to the rules of the journal.
Response 29: Thanks for your suggestion.
Format of References have been redesigned according to the rules of the journal.

Reviewer 4 Report
Comments and Suggestions for Authors
The study is of undoubted interest for studying the biosynthesis of agar and for the selection of agar strains, as well as optimizing the cultivation conditions of Gracilariopsis lemaneiformis. Here are some notes.
Remove the word title from the title of the manuscript.
It is advisable to expand the title of the article.
Provide a photo of the object being studied.
Describe the purpose of the study specifically.
In the Materials and Methods section, remove the first paragraph with the requirements for the description of the section.
Make corrections to Table 3 in the Materials and Methods section.
The salt concentration should be given in mM.
Give a histological picture of the localization of agar in the cell. (Inner matrix of cell walls).
It is necessary to bring the list of references in accordance with the requirements of the journal.

Author Response
Thank you for your suggestion. The corresponding information has been added to the following document.

Round 2
Reviewer 1 Report
Comments and Suggestions for Authors
The authors introduced the corrections suggested by the reviewers, so the manuscript can now be accepted for publication
Author Response
Thanks!
Reviewer 2 Report
Comments and Suggestions for Authors
No reply to reviewer's comments have been received.
In the previous round of the review was mentioned:
"The paper of Li et al. entitled “Identification of Key genes for Agar Biosynthesis in Gracilariopsis lemaneiformis (Rhodophyta) under Different Treatments” fails to convince that the findings can be used as an indicator of agar accumulation in this red alga. First of all, looking at gene expression is not the best tool to understand what is going on during different treatments, that is, salinity, temperature, and nutrients. Transcriptomics, in general, is a poor indicator of complex biochemical processes. Proteomics and metabolomics provide a much better understanding of these processes.
The pattern of expression of the 11 genes is very different when looking at various salinity, temperature, nitrogen, and phosphorous conditions. The idea of a relevant correlation is not clear from the data. The composite analyses from Table 1 and Figure 5 are not convincing. If a strong positive correlation was found between gpi, mpi galt and mpg on day 9, what about the other days? Quite interestingly, day 9 is also the last time point! Theoretically, the experimental design was not OK because it is essential to know if this trend is maintained or not after day 9!
This is one of the multiple issues with this paper. The usage of the English language is rather poor, that is, there are many instances in which the text has to be revised, misspelling can be found everywhere in the text, including in figures (for example, Figure 3 - Cltivation time instead of cultivation time), the text and especially references were not well formatted.
To conclude, the topic was not addressed using the correct approach, that is, transcriptomics was used instead of proteomics and metabolomics. The experimental design is not OK, as data collection stopped on day 9, the only one in which some relevant data is suggested. Because of the problems with the method used, and with the experimental design, the conclusions of this work are not sustainable.
Comments on the Quality of English LanguageModerate editing of English language required
Author Response

(The authors gave the same response as above.)

Reviewer 3 Report
Comments and Suggestions for Authors
The abstract does not meet the requirements of the journal in the section “1) Background: Place the question addressed in a broad context and highlight the purpose of the study;”.
Lines 13–14: “Both high and low salinity promoted agar accumulation in G. lemaneiformis.” Add numbers (%) to demonstrate the agar increase.
Lines 14–15: “The agar accumulation was inversely proportional to the change in relative growth rate of the algae.” At 40% salt, the algae growth rate was low, but at 10% salt, the algae growth rate was high (Figure 1A). At the same time, the agar content increased in both variants (Figure 2A).
Lines 456–457: “adding a certain concentration of nitrogen and phosphorus into seawater is beneficial for algal growth in the laboratory.” - This work does not show the positive effect of adding phosphorus for algal growth in the laboratory.
In Table 1, the authors provided the values of “changes in agar content (%)” for different treatments at the beginning of the experiment and on day 15. In a previous review, I asked the authors to provide results on agar production (accumulation) under different conditions. How many grams of agar could the algae produce under different conditions? How did the yield (not content, %) of agar change at different salinity, temperature, nitrogen and phosphorus concentrations? These data are important for assessing the biotechnological prospects for applying the results presented by the authors. For example, 5 grams of algae with a 20% agar content have 1 gram of agar. After 15 days of growth, the mass of the algae increased to 5.5 grams. When maintaining a 20% agar content, the algae contains 1.1 grams of agar. Therefore, 0.1 gram of agar was produced in 15 days.
Figure 6: The figure shows the incorrect formulas “GDP-L-galactose” and “Agar precursor”.
Tables 4 and 5: Use superscripts and subscripts when writing chemical formulas and numbers.
Line 2237: Add an explanation to the manuscript of what is “weight of soluble polysaccharides”.
Author Response

(The authors gave the same response as above.)

Round 3
Reviewer 2 Report
Comments and Suggestions for Authors
I am sorry, but the topic was not addressed using the correct approach, that is, transcriptomics was used instead of proteomics and metabolomics. Metabolomics would be the best approach.
The experimental design is not OK, as data collection stopped on day 9, the only one in which some relevant data is suggested. What trends were observed after day 9? "In general, under successive induction conditions, the physiology and biochemistry of the cells was initially drastically altered, but they gradually adapt to the environment and reach a stable state." What is the experimental evidence for this?
Also, what "According to table S1, the expression trend of genes gradually concentrates in 1-9 days of treatments." means?
Unfortunately, the conclusions of this work are not sustainable.
Comments on the Quality of English LanguageModerate editing of English language required
Reviewer 3 Report
Comments and Suggestions for Authors
I see some inconsistencies between the text of the manuscript and the authors' responses. The main idea in the manuscript is the study of the mechanisms of agar biosynthesis, but the biotechnological conclusions are bypassed. In their responses, the authors write that in their work they were interested in the activity of agar biosynthesis genes outside the context of the agar harvest. But even in the aspect of studying genes, the authors do not give the answer given by the title of the manuscript. Which genes are key for agar biosynthesis under different conditions? The gpiI, gpiII, mpi, galt, and mpg genes identified by the authors in cells may be involved in other metabolic pathways besides agar biosynthesis (especially gpiI and gpiII, which are part of the glycolysis genes). This issue was not discussed by the authors. Options where the authors observed an increase in agar production, but there was no increase in the activity of these genes, were also not discussed. The activity of gme, which involves in the biosynthesis of the key metabolite in agar synthesis, GMP-L-galactose, does not correlate with agar content. This is also not discussed by the authors.
In this regard, I think that the authors should not limit themselves to minor corrections of the manuscript, but should thoroughly revise it so that the discussion and conclusions are consistent with the results and purpose of this work.
Figure 6 could be improved further. For example, why do the authors use “GMP” for the enzyme encoded by the mpg gene? For all other enzymes, the names are the same as the names of the genes. In addition, the agar precursor formula contains numbers, but nowhere is it explained what they mean. And since the agar precursor consists of residues of different monosaccharides, it would be more correct to use different numbering: 1-6 and 1’-6’.